# Study of Spatiotemporal Changes and Driving Factors of Habitat Quality: A Case Study of the Agro-Pastoral Ecotone in Northern Shaanxi, China

**Guoyi Cui [1], Yan Zhang [1,2,3,*], Feihang Shi [1], Wenxia Jia [1], Bohua Pan [1], Changkun Han [1], Zhengze Liu [1], Min Li [1] and Haohao Zhou [1]**

[1] School of Land Engineering, Chang'an University, Xi'an 710054, China; 2019127078@chd.edu.cn (G.C.); 2019127069@chd.edu.cn (F.S.); 2020135027@chd.edu.cn (W.J.); 2020135033@chd.edu.cn (B.P.); 2020135015@chd.edu.cn (C.H.); 2020235007@chd.edu.cn (Z.L.); 2020135017@chd.edu.cn (M.L.); 2020135009@chd.edu.cn (H.Z.)
[2] Shaanxi Key Laboratory of Land Consolidation, Xi'an 710054, China
[3] Key Laboratory of Degraded and Unused Land Consolidation Engineering, Ministry of Natural Resources, Xi'an 710075, China
\* Correspondence: zyzhangyan@chd.edu.cn

**Abstract:** Habitat quality is a key indicator for assessing the biodiversity-maintenance functions of ecosystem services. The issue of habitat quality changes in semi-arid and arid areas has been becoming serious, but there are few deep investigations of habitat quality in these regions, such as studies of the temporal and spatial changes of habitat quality and its driving forces. This study focuses on the agro-pastoral ecotone of northern Shaanxi with vulnerable biodiversity. By using the Fragstats software, the InVEST model, and the Geo-detector model, we analyzed land-use data collected from 1990, 2000, 2010, and 2020, and we explored the landscape pattern index, the spatial and temporal variation of habitat quality, and the influence of its drivers. GDP, population density, precipitation, temperature, land use, NDVI, elevation, and slope were detected by Geo-detector. The research results show that: (1) Arable land and grassland were the dominant land types from 1990 to 2020, and there was significant mutual circulation between arable land and grassland. Forest area increased by 24%. Many other land-use types were transformed into construction land, and construction land increased by 727% compared with the base period. (2) Landscape heterogeneity increased in the study region, shown by the fractured structure of the overall landscape and by the aggravated human disturbance of the landscape. (3) Average habitat quality underwent a trend of oscillation. Regarding spatial distribution, habitat quality was higher in the east than in the west. (4) The influencing factors of habitat quality monitored by Geo-detectors show that the driving force of land use on habitat quality was the strongest, followed by precipitation and vegetation coverage. Elevation, slope, GDP, and population density had the least influence on habitat quality. The bi-factor interaction enhanced habitat quality to different levels. This study is critical to the conservation of biodiversity and to ecological civilization construction in arid and semi-arid regions.

**Keywords:** habitat quality; InVEST model; land-use change; spatiotemporal change; geo-detector

## 1. Introduction

The ability of ecosystems to provide suitable conditions for species survival and development is referred to as habitat quality (HQ) [1,2]. Habitat quality can reflect the maintenance function of the biodiversity of ecosystems [3]. With the advancement of industrialization and urbanization, human activities have an increasingly significant impact on land-use patterns and landscape patterns. Land is the carrier of habitat. Human activities change the situation of land cover, leading to changes in habitat quality [4–6].

The quality of habitats is critical to the long-term development of humans and other species. Generally, the methods for assessing habitat quality fall into two categories: (1) Establish an index evaluation system to obtain suitable habitat parameters for specific species through field surveys, and build a habitat quality evaluation system [7,8]. This method is only suitable for the study of small-scale areas, because it requires lots of labor and time and a high overall operation cost. (2) Use the model evaluation method, such as the HIS model [9], the social value ecosystem service model (SoIVES) [10], and the Integrated Valuation of Ecosystem Services and Tradeoffs (InVEST) [11,12]. Among these models, the InVEST model is often used as an ecosystem service assessment tool to calculate spatial effects. The method of the habitat quality module in the InVEST model to assess habitats is to establish a relationship between different land-use types and threat sources by using the land-use type data of the study area and by combining the habitat suitability, habitat sensitivity, and threat intensity of each land-use type [13]. This model has the advantages of spatial mapping, easy data acquisition, and high evaluation accuracy [14]. Currently, the module is widely used. For instance, Yang used the InVEST habitat quality module to evaluate the temporal and spatial changes of habitat quality in the Taihang Mountains from 1990 to 2020 and the response of land use to habitat quality [15]. Yohannes et al. explored the relationship between habitat quality and landscape characteristics in the Beressa watershed from 1972 to 2047, as well as temporal and geographical variations in habitat quality [16]. Upadhaya et al. evaluated land-use changes in the Alabaha River Basin in southeastern Georgia and its impact on habitat quality [17]. Zhu et al. used the InVEST habitat quality module to assess the habitat quality of Hangzhou from 2004 to 2015, and they investigated the effects of urbanization and landscape pattern changes on habitat quality using ordinary least squares and geographic weighted regression models [18].

For research methodology, most studies have focused on spatial and temporal evolution and on the prediction of habitat quality. The methods of influencing factors affecting habitat quality mainly use correlation analysis [16] and geographical weighted regression models (GWR) [15], but these methods cannot explore interactions among multiple factors on habitat quality. Geo-detectors can detect both numerical data and qualitative data, and they can also detect interactions between the two factors with the dependent variable. However, there are still relatively few studies on habitat quality using Geo-detectors. Most studies have focused on mountains, watersheds, and administrative regions. They have also investigated these areas' temporal and spatial changes in habitat quality as well as the relationship between land use and habitat quality. However, the issue of habitat quality in arid and semi-arid regions has attracted increased attention [19–22]. The agro-pastoral ecotone of northern Shaanxi is a transitional zone between semi-arid and arid areas, with a low environmental carrying capacity and a fragile ecological environment [23–25]. The agro-pastoral ecotone in northern Shaanxi is also an important coal mining area in China. With excessive agricultural planting, urban development, and industrial mining production activities in recent decades, land-use types and landscape patterns have changed. Local ecological environments have undergone enormous pressure [26]. Therefore, this study analyzes changes in land use and landscape pattern indexes in the agro-pastoral ecotone of northern Shaanxi, using the InVEST model to study the temporal and spatial changes of habitat quality as well as the factors that drive them. The research objectives of this study are as follows: (1) to study the changes in land use and landscape patterns from 1990 to 2020; (2) to study the spatiotemporal changes of habitat quality; and (3) to detect the driving forces of factors affecting habitat quality. This study provides a theoretical basis for biodiversity conservation, nature reserve planning, and ecological civilization construction in the agricultural and animal husbandry interlaced areas in northern Shaanxi.

## 2. Materials and Methods

### 2.1. Study Area

The agro-pastoral ecotone in northern Shaanxi is in the northernmost part of the Shaanxi Province in China, covering two districts and five counties (Yuyang District, Hengshan District, Fugu County, Jingbian County, Jia County, Shenmu County, Dingbian County) in the northern part of Yulin City (Figure 1). The geographical coordinates of the study area are between 107°28′~111°15′ E and 36°57′~39°34′ N. The region is in the middle reaches of the Yellow River, belonging to the junction of the Loess Plateau and Mu Us Sand Land and bordering four provinces and autonomous regions: Shanxi, Inner Mongolia, Ningxia and Gansu. The study area has a temperate semi-arid climate with an annual mean temperature of 8.3 °C and significant temperature fluctuations throughout the year. The average annual precipitation is 365.7 mm, with little precipitation and large evaporation, gradually increasing from northwest to southeast. Summer is when most of the annual precipitation falls. The main natural vegetation is desert grassland and shrubs.

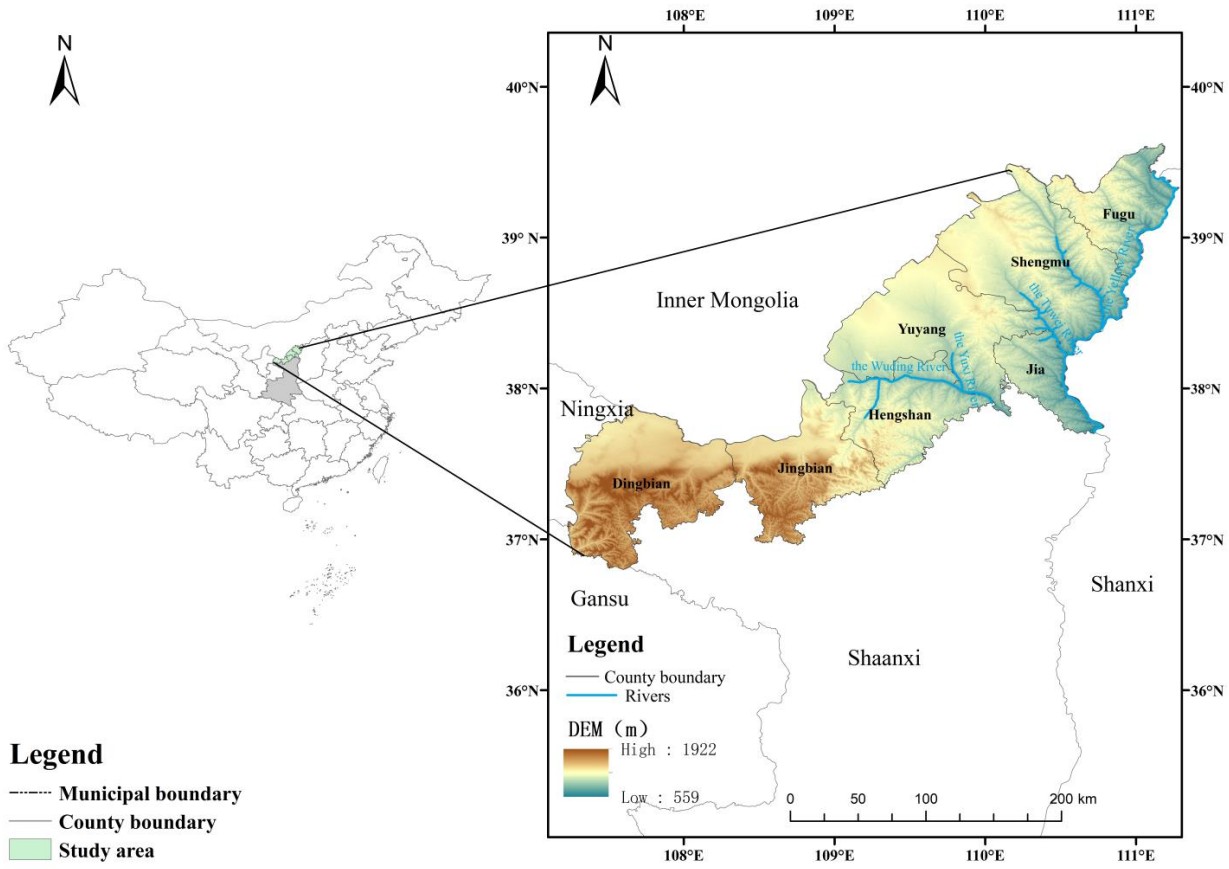

**Figure 1.** Map of study area.

There are abundant mineral resources such as coal and petroleum in the agro-pastoral ecotone of northern Shaanxi, which serves as a national-level energy and a chemical base for coal, electric power, oil and gas, and chemical industries. The local habitat quality has deteriorated because of the continuous expansion of urbanization and industrialization.

### 2.2. Data Source

The land-use data with a resolution of 30 m from 1990 to 2020 used in this study was obtained by visual interpretation of Landsat 8 images, including 6 primary land types and 20 secondary land types (Table 1). The average annual precipitation, the average annual temperature, NDVI, GDP, and population density raster data resolution are all 1 km$^2$. The above data are all from the Resource and Environmental Science Data Center

of the Chinese Academy of Sciences (http://www.resdc.cn (accessed on 15 July 2021)). Specifically, temperature distribution raster data and precipitation distribution raster data were generated using the ANUSPLIN interpolation software. The NDVI spatial-distribution dataset was based on SPOT/VEGETATION PROBA-V 1 KM PRODUCTS ten-day 1 km vegetation index data, based on monthly data using the maximum synthesis method to generate annual vegetation index datasets since 1998. GDP spatial-distribution raster data are based on the national GDP statistics data of counties and comprehensively consider multiple factors, such as land-use type, night light brightness, and the density of settlements that are closely related to human economic activities. This method spreads the GDP data, with the administrative area as the basic statistical unit, to the grid unit to realize the spatialization of the GDP. Road data from 2018 include railways, highways, national roads, provincial roads, and county roads, which was obtained from the National Geomatics Center of China (www.ngcc.cn (accessed on 31 July 2021)). DEM data with a resolution of 90 m were downloaded in the geospatial data cloud (https://www.gscloud.cn (accessed on 30 July 2021)). The slope data were obtained from the DEM data.

**Table 1.** Land-use classification system.

| Primary Land-Use Type | Secondary Land-Use Type |
| --- | --- |
| Arable land | Paddy field, Dry land |
| Forestland | Forest, Shrub, Sparse Forest, Other forest |
| Grassland | High-coverage grassland, Moderate-coverage grassland, Low-coverage grassland |
| Water | River/Canal, Lake, Reservoir/Pond, Mudflat, Wetland |
| Construction land | Urban, Residents, Other construction |
| Unused land | Sand land, Salinity land, Bare land |

*2.3. Methods*

2.3.1. Analysis of the Landscape Pattern Index

Landscape pattern usually refers to the spatial structure characteristics of the landscape, specifically referring to the formation of natural or artificial characteristics, a series of characteristics with different sizes and shapes, or the arrangement of different landscape mosaics in the landscape space. Fragstats is a spatial pattern tool used for quantifying the structure of landscapes. It is used to calculate metrics of landscapes [27]. This software was developed by the University of Massachusetts in 1995. In this study, the landscape pattern index is evaluated using the Fragstats (version 4.2) software. Based on the characteristics of the study area, 11 landscape pattern indexes were finally selected: Patch Density, Largest Patch Index, Average Patch Area, Aggregation Index, Edge Density, Area-Weighted Mean Fractal Dimension Index, Contagion Index, Landscape Shape Index, Interspersion and Juxtaposition Index, Shannon's Evenness Index (SHEI), and Shannon's Diversity Index (see Table 2).

**Table 2.** Description of the Landscape Index.

| Metric | Acronym | Scale | Description |
|---|---|---|---|
| Patch Density | PD | Class/ Landscape | This index represents the number of patches per 100 ha. It reflects the degree of fragmentation of the landscape. The larger that the PD value is, the more fragmented that the landscape is. |
| Mean of Patch Area | AREA_MN | Class/ Landscape | This index indicates the extent of landscape fragmentation. The smaller that the value is, the more broken that it is. |
| Edge Density, | ED | Class/ Landscape | Edge density includes both landscape edge density and the type of edge density. Landscape edge density refers to the edge length between landscape patches per unit of area within the landscape range. The type of edge density refers to the edge length of a type of landscape patch with an adjacent patch. |
| Largest Patch Index | LPI | Class/ Landscape | The Largest Patch Index is used to determine the dominant patch type in the landscape. |
| Landscape Shape Index | LSI | Class/ Landscape | Landscape Shape Index indicates the shape index of patches in a landscape pattern. |
| Area-Weighted Mean Fractal Dimension Index | FRAC_AM | Class/ Landscape | This index reflects the degree of irregularity and fragmentation of the landscape type. The more irregular that the shape of the landscape is, the higher that the value of the fractal dimension is. Values range from 1 to 2, and values closer to 1 indicate areas of relatively simple shapes, such as a square or a circle. Values close to 2 indicate both complex and irregular shapes. |
| Shannon's Diversity Index | SHDI | Class/ Landscape | This index can reflect landscape heterogeneity and is particularly sensitive to uneven distributions of different patchwork types in a landscape. SHDI is also a sensitive indicator when comparing and analyzing the diversity and heterogeneity of different landscapes or of the same landscape in different periods. |
| Shannon's Evenness Index | SHEI | Class/ Landscape | SHEI equals the Shannon Diversity Index divided by the maximum possible diversity at a given landscape abundance (equal distribution across patch types). |
| Interspersion and Juxtaposition Index | IJI | Class/ Landscape | IJI is one of the most important indicators to describe the spatial pattern of a landscape. IJI has a significant response to the distribution of characteristics of ecosystems that are severely constrained by certain natural conditions. |
| Contagion Index, | CONTAG | Landscape | This index describes the degree of non-randomness or clustering of different patch types in the landscape. High values indicate good connectivity of a dominant patch type in the landscape; low values indicate a dense pattern with multiple elements and a high degree of landscape fragmentation. |
| Aggregation Index | AI | Class/ Landscape | AI represents connectivity between patches of each landscape type. |

2.3.2. Assessment of Habitat Quality

According to the habitat quality module of the InVEST model, locations with good habitat quality also have a significant biodiversity maintenance function. The InVEST

model finally generates a habitat quality map, showing a range of habitat quality values between 0 and 1. The calculation formula is as follows [28]:

$$D_{xj} = \sum_{r=1}^{R} \sum_{y=1}^{Y_r} r_y \left( \frac{\omega_r}{\sum_{r=1}^{R} \omega_r} \right) i_{rxy} \beta_x S_{jr} \tag{1}$$

$$i_{rxy} = 1 - \left( \frac{d_{xy}}{d_{rmax}} \right) (if\ linear) \tag{2}$$

$$i_{rxy} = \exp \left( \frac{-2.99 d_{xy}}{d_{rmax}} \right) (if\ exponential) \tag{3}$$

where $D_{xj}$ denotes the degree of habitat degradation in grid cell $x$ with habitat type $j$; $R$ denotes the number of potential threats; $Y_r$ is the grid number of $r$ on a raster map; $r_y$ is the intensity of grid cell $y$; $\omega_r$ denotes the threat source's weight; and the distance between the habitat and the threat source is indicated by irxy. $\beta_x$ means the anti-interference level of the grid cell $x$; $S_{jr}$ denotes the relative sensitivity of habitat type $j$ to the threat source $r$; $d_{xy}$ is the distance between grid cells $x$ and $y$; and drmax is the maximum impact distance of the threat source $r$.

$$Q_{xj} = H_j \left[ 1 - \left( \frac{D_{xj}^2}{D_{xj}^2 + K^2} \right) \right] \tag{4}$$

where $Q_{xj}$ represents the habitat quality of pixel $x$ in land-use/cover type $j$; $D_{xj}$ represents the threat level of pixel $x$ in land-use/cover type $j$; $H_j$ represents the habitat appropriateness of land-use/cover type $j$; and $K$ represents half of the saturation constant (which is half of the maximum value of $D_{xj}$). The model's default parameter, $Z$, is set to 2.5.

The habitat quality module needs data, including land-use type data, habitat threat factors, threat source factor weights, influence distances, and the landscape types' sensitivity to threat sources [16]. This study is based on the InVEST model manual [29] and on previous studies [30–35], combined with the actual situation of the study area to determine the relevant parameters and to design the habitat quality module input parameter table (Tables 3 and 4).

**Table 3.** The threat source and related coefficients.

| Threat Source | Weight | Maximum Impact Distance (Km) | Decay Type |
|---|---|---|---|
| Arable land | 0.6 | 3 | linear |
| Urban | 1 | 10 | exponential |
| Residents | 0.7 | 2 | exponential |
| Other construction land | 0.5 | 1 | exponential |
| Highway | 0.6 | 3 | linear |
| National road | 0.6 | 3 | linear |
| Provincial road | 0.5 | 2 | linear |
| County road | 0.4 | 1 | linear |
| Railway | 1 | 5 | linear |

**Table 4.** The sensitivity of land-use types to each threat source.

| Land-Use Type | | Habitat Suitability | Threats | | | | | | | | |
| Primary Land-Use Types | Secondary Land-Use Type | | Arable Land | Urban | Rural Residents | Construction Land | Highway | National Road | Provincial Road | County Road | Rail-Way |
|---|---|---|---|---|---|---|---|---|---|---|---|
| Arable Land | Paddy field | 0.5 | 0.3 | 0.4 | 0.3 | 0.4 | 0.7 | 0.6 | 0.4 | 0.3 | 0.7 |
| | Dry land | 0.5 | 0.3 | 0.4 | 0.3 | 0.4 | 0.7 | 0.6 | 0.4 | 0.3 | 0.7 |
| Forestland | Woodland | 1 | 0.5 | 0.8 | 0.7 | 0.8 | 0.7 | 0.6 | 0.5 | 0.4 | 0.8 |
| | Shrub | 0.8 | 0.5 | 0.7 | 0.6 | 0.7 | 0.7 | 0.6 | 0.5 | 0.4 | 0.8 |
| | Sparse forest | 0.5 | 0.4 | 0.5 | 0.4 | 0.55 | 0.7 | 0.6 | 0.5 | 0.4 | 0.8 |
| | Other forest | 0.3 | 0.2 | 0.2 | 0.2 | 0.45 | 0.7 | 0.6 | 0.5 | 0.4 | 0.8 |
| Grassland | High-coverage grassland | 0.8 | 0.4 | 0.65 | 0.6 | 0.75 | 0.5 | 0.4 | 0.3 | 0.2 | 0.55 |
| | Moderate-coverage grassland | 0.5 | 0.3 | 0.55 | 0.5 | 0.6 | 0.5 | 0.4 | 0.3 | 0.2 | 0.55 |
| | Low-coverage grassland | 0.3 | 0.2 | 0.5 | 0.4 | 0.55 | 0.5 | 0.4 | 0.3 | 0.2 | 0.55 |
| Water | River | 0.8 | 0.65 | 0.8 | 0.7 | 0.9 | 0.65 | 0.6 | 0.5 | 0.3 | 0.7 |
| | Lake | 0.9 | 0.65 | 0.8 | 0.7 | 0.9 | 0.65 | 0.6 | 0.5 | 0.3 | 0.7 |
| | Pond | 0.8 | 0.65 | 0.8 | 0.7 | 0.9 | 0.65 | 0.6 | 0.5 | 0.3 | 0.7 |
| | Beach | 0.5 | 0.2 | 0.3 | 0.2 | 0.3 | 0.65 | 0.6 | 0.5 | 0.3 | 0.7 |
| | Marsh | 0.9 | 0.65 | 0.7 | 0.6 | 0.7 | 0.65 | 0.6 | 0.5 | 0.3 | 0.7 |
| Construction Land | Urban | 0 | 0 | 0 | 0 | 0 | 0 | 0 | 0 | 0 | 0 |
| | Rural residents | 0 | 0 | 0 | 0 | 0 | 0 | 0 | 0 | 0 | 0 |
| | Other construction land | 0 | 0 | 0 | 0 | 0 | 0 | 0 | 0 | 0 | 0 |
| Unused Land | Sand land | 0.1 | 0 | 0 | 0 | 0 | 0.2 | 0.1 | 0.1 | 0.1 | 0.2 |
| | Salinity land | 0.2 | 0.1 | 0.3 | 0.2 | 0.3 | 0.1 | 0.2 | 0.1 | 0.1 | 0.2 |
| | Bare land | 0 | 0 | 0 | 0 | 0 | 0 | 0 | 0 | 0 | 0 |

### 2.3.3. Global Moran's I

Global Moran's I, a measure of spatial autocorrelation proposed by Patrick Moran, is designed to describe the extent of habitat quality across the study area. The global Moran index takes the value interval as $[-1,1]$. The formula is as follows:

$$I = \frac{n \sum\limits_{i=1}^{n} \sum\limits_{j=1}^{n} w_{ij}(y_i - \overline{y})(y_j - \overline{y})}{\sum\limits_{i=1}^{n} (y_i - \overline{y})^2} \tag{5}$$

$$Z_{score} = \frac{I - E(I)}{\sqrt{VAR(I)}} \tag{6}$$

where $I$ is the global Moran index; $n$ is the total number of study units; $\overline{y}$ is the average of the habitat quality; $y_i$ and $y_j$ are the $i$ and $j$ study area habitat quality; $w_{ij}$ is the spatial weight coefficient of regions $i$ and $j$, which reflects the spatial relationship of regions $i$ and $j$ and is defined as $w_{ij} = 1$ and otherwise as $w_{ij} = 0$; and $E(I)$ and $VAR(I)$ represent the expected value and variance of the Moran index $y$, respectively.

### 2.3.4. Hot-Spot Analysis (Getis-Ord Gi*)

Hot-spot analysis is widely used in geographic studies, in which hot spots and cold spots represent high-value aggregation and low-value aggregation of geographic elements in space, respectively. The formula is as follows:

$$G = \left(\sum_{j=1}^{n} w_{i,j} y_j - \overline{y} \sum_{j=1}^{n} w_{i,j}\right) / {}^{s}\sqrt{n\left(\sum_{j=1}^{n} w^2{}_{i,j} - \left(\sum_{j=1}^{n} w_{i,j}\right)^2\right) / (n-1)} \tag{7}$$

where $G$ is the local hotspot analysis value; $y_j$ is the attribute value of element $j$; $w_{i,j}$ represents the spatial weight between elements $i$ and $j$ (adjacent 1, no adjacent 0); $S$ is the standard deviation of the output in the study period; and n is the total number of sample points. The G statistical result is the Z score. The Z score is positive and significant, indicating hot spots. The larger the value, the more closely the hot spots are clustered; the negative and significant Z score means the cold spots, and the smaller the value, the closer the cold spots are clustered.

### 2.3.5. Geo-Detector

Geo-detector is a tool for detecting spatial heterogeneity [36]. Geo-detectors are divided into four types, namely factor detection, ecological detection, interaction detection, and risk detection. They are widely used to detect the driving factors of geographical things, especially in the field of ecological research [37–39]. In this study, the driving factors of the spatial differentiation of habitat quality in the study area were investigated using single-factor detection and interaction detection. Among them, the single-factor detector is primarily used to analyze the level of the driving force of each impact factor on habitat quality, and the formula is as follows: [36]:

$$q = 1 - \frac{\sum\limits_{i=1}^{m} n_i \sigma_i^2}{n\sigma^2} = 1 - \frac{SSW}{SST}$$
$$SSW = \sum\limits_{i=1}^{m} n_i \sigma_i^2,$$
$$SST = n\sigma^2 \tag{8}$$

where $i$ ($i = 1, 2, \ldots, l$) is the stratification of dependent variable y or independent variable %, i.e., the classification or partition; $n_i$ and $n$ are the unit numbers of layer $i$ and the whole region, respectively; $\sigma^2{}_i$ and $\sigma^2$ are the variances of layer i and region Y, respectively; and SSW and SST are the sum of variances within the layer and the total variances of the

whole region, respectively. The value range of *q* is [0,1], where the larger that the value is, the stronger that the explanatory power of independent variable % is to dependent variable *y*.

Interaction detection is mainly used to determine whether multiple influence factors increase or decrease the driving power of habitat quality after interaction, by comparing the q-values of single factors (q (X1) and paired factors (q (X1 ∩ X2)). Table 5 [40] lists the main types.

**Table 5.** Types of interactions between two factors.

| Criterion | Interactive Forms |
| --- | --- |
| P (X1 ∩ X2) < Min[P(X1), P(X2)] | Weakened, nonlinear |
| Min[P(X1), P(X2)] < P (X1 ∩ X2) < Max[P(X1), P(X2)] | Weakened, single-factor nonlinear |
| P (X1 ∩ X2) > Max[P(X1), P(X2)] | Enhanced, double factors |
| P (X1 ∩ X2) > P(X1) + P(X2) | Enhanced, nonlinear |
| P(X1 ∩ X2) = P(X1) + P(X2) | Independent |

Note: "P(X1 ∩ X2)" refers to the interaction of paired factors.

This study selected population density, GDP, annual precipitation, annual temperature, land-use type, Normalized Difference Vegetation Index (NDVI), slope, and elevation as detection factors (Figure 2). The grid was used to extract the numerical value of each factor layer, discretize it, and input it into the Geo-detector model to run.

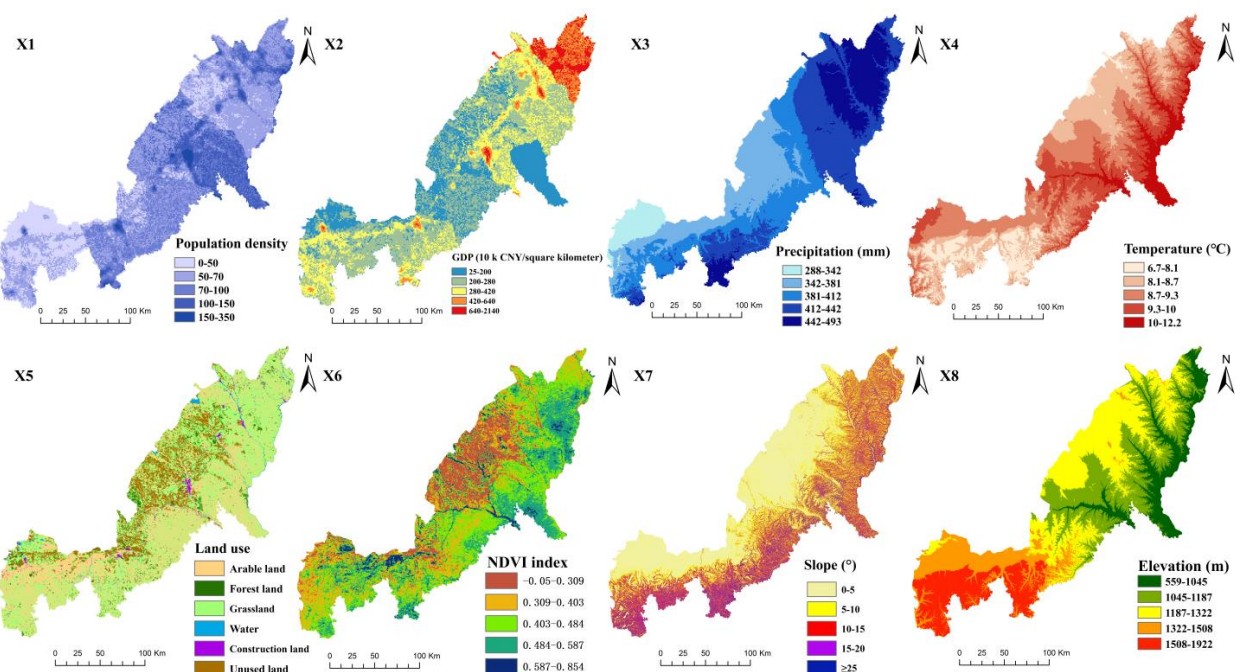

**Figure 2.** Stratification of influencing factors driving habitat quality. Note: X1: population density; X2: GDP 10 thousand per km$^2$; X3: annual precipitation; X4: annual temperature; X5: land-use type; X6: NDVI X7 slope; X8: elevation.

## 3. Results

### 3.1. Land-Use Change in the Agro-Pastoral Ecotone of Northern Shaanxi

According to land-use data from 1990, 2000, 2010, and 2020, the main types of land use in study area are grassland and arable land, accounting for 80%. Arable land decreased by 477.2 km$^2$ (4%) from 1990 to 2010 and slightly increased by 2.47 km$^2$ (0.02%) from 2010 to 2020, whereas grassland increased by 1442.26 km$^2$ (9.5%) from 1990 to 2010 and decreased by 474.6 km$^2$ (2.8%) from 2010 to 2020. In the past 30 years, water and unused land

decreased by 40.04 km$^2$ (8.6%) and 1600.21 km$^2$ (27%), respectively, whereas forestland increased by 271.3 km$^2$ (23%). Construction land increased from 128 km$^2$ in 1990 to 1058.04 km$^2$ in 2020, of which the area increased significantly by 930.04 Km$^2$ (727%) and gradually gathered as a band-like distribution (Figure 3).

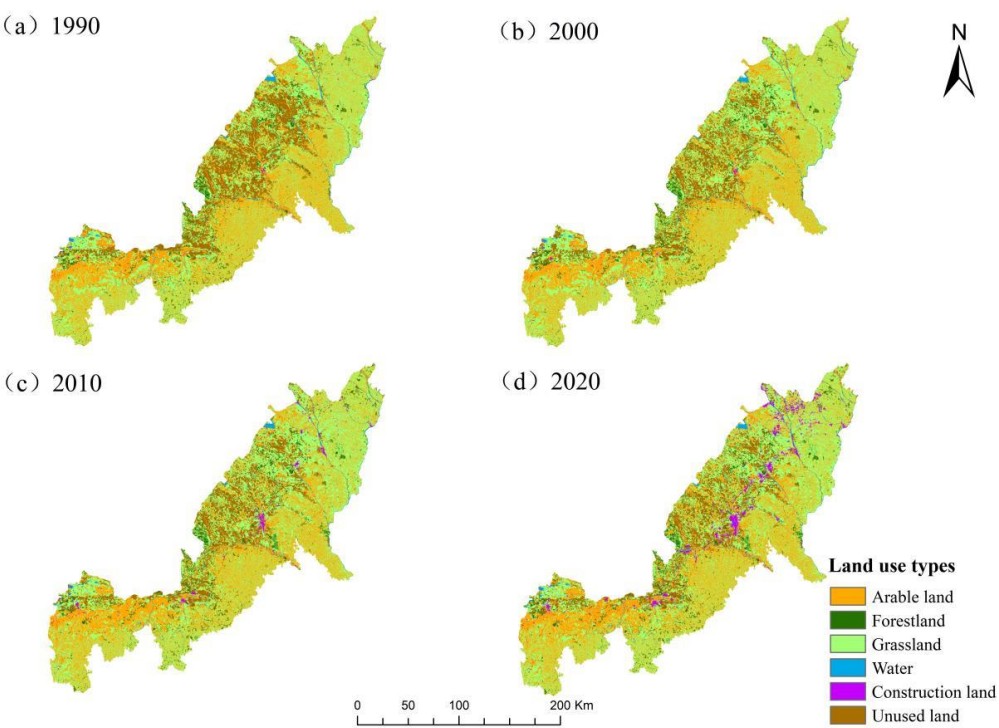

**Figure 3.** Land-use types in the agro-pastoral ecotone of northern Shaanxi from 1990 to 2020: (**a**) 1990, (**b**) 2000, (**c**) 2010, (**d**) 2020.

The change in each land-use type is shown based on the analysis of the land-use transfer matrix (Table 6). In the past 30 years, arable land has been mainly transformed into grassland and forestland. The area of arable land that changed to grassland was 1699.65 km$^2$ between 2000 and 2020, whereas the area of arable land that converted to forestland was 399.17 km$^2$, with significant variations. The main flow direction of forestland was grassland, with an outflow area of about 97.94 km$^2$, which accounts for 39% of the reduced forestland. In addition, from 2000 to 2010, 19.16 km$^2$ of forestland was converted into unused land. From 2010 to 2020, 52.33 km$^2$ of forestland was transferred to arable land, and 31.22 km$^2$ of forestland was converted to construction land. The main land types converted by grassland are arable land, construction land, and unused land, and the circulation areas are 1471.47 km$^2$, 472.29 km$^2$, and 261.25 km$^2$, respectively. Among them, changes during the period of 2000–2020 were dramatic. Water area was primarily converted into arable land and grassland, particularly between 2000 and 2020, accounting for approximately 76% of the reduced water area. The area of construction land that was transferred to other land-use types was 102.97 km$^2$. On the other hand, other land types were turned into development land in huge numbers. From 2000 to 2020, due to a large reduction in grassland, arable land, and unused land, construction land continued to grow. It accounted for 94% of the increased construction land area. From 1990 to 2010, the main flow of unused land was grassland, which accounted for 95% of the reduced unused land area. From 2010 to 2020, the unused land flows were mainly arable land, construction land, and grassland. This was related to the local implementation of land consolidation projects [41].

**Table 6.** Land-use type change matrix from 1990 to 2020 (km$^2$).

| Time Interval | Land-Use Type | Arable Land | Forestland | Grassland | Water | Construction Land | Unused Land |
|---|---|---|---|---|---|---|---|
| | Arable land | 12,727.38 | 13.83 | 25.09 | 1.52 | 10.99 | 34.84 |
| | Forestland | 3.21 | 1125.36 | 24.19 | 0.00 | 0.24 | 1.95 |
| | Grassland | 39.52 | 27.62 | 14,966.03 | 2.21 | 2.08 | 76.76 |
| 1990–2000 | Water | 4.66 | 0.31 | 7.68 | 446.32 | 0.00 | 2.81 |
| | Construction land | 0.01 | 0.00 | 0.00 | 0.00 | 127.98 | 0.00 |
| | Unused land | 30.91 | 36.09 | 1294.49 | 0.12 | 2.79 | 4459.26 |
| | Arable land | 11,612.49 | 207.36 | 890.23 | 8.90 | 78.55 | 9.64 |
| | Forestland | 21.2 | 1138.85 | 15.53 | 0.49 | 8.29 | 19.16 |
| | Grassland | 590.79 | 114.11 | 15,438.16 | 8.79 | 96.31 | 73.49 |
| 2000–2010 | Water | 20.97 | 1.1 | 21.65 | 401.44 | 5.85 | 0.94 |
| | Construction land | 25.88 | 0.3 | 3.76 | 0.99 | 113.13 | 0.04 |
| | Unused land | 65.13 | 5.08 | 187.11 | 4.84 | 59.47 | 4255.7 |
| | Arable land | 11,231.8 | 43.53 | 789.42 | 29.71 | 191.81 | 48.88 |
| | Forestland | 52.33 | 1310.95 | 58.22 | 2.13 | 31.22 | 11.55 |
| | Grassland | 841.16 | 57.74 | 15,063.44 | 29.45 | 375.98 | 184.49 |
| 2010–2020 | Water | 22.64 | 0.89 | 28.09 | 351.62 | 16.56 | 3.81 |
| | Construction land | 28.12 | 2.03 | 19.56 | 8.37 | 297.03 | 6.43 |
| | Unused land | 162.88 | 11.13 | 123.13 | 6.47 | 145.44 | 3908.27 |

*3.2. Analysis of Landscape Pattern Index in the Agro-Pastoral Ecotone of Northern Shaanxi*

At the landscape level (Table 7), the Patch Density of the study area increased by 25% to 2020, but Average Patch Area decreased by 20%. Edge Density and the Landscape Shape Index showed a trend of first increasing, then decreasing, and then increasing, with a significant growth trend from 2010 to 2020, increasing by 4% and 5%, respectively. The Area-Weighted Mean Fractal Dimension Index and Aggregation Index both showed a decreasing, increasing, and then decreasing trend. The Interspersion and Juxtaposition Index, Shannon's Diversity Index, and Shannon's Evenness Index continued to increase after decreasing to their minimum values between 1990 and 2000. The Largest Patch Index continued to increase by 7% from 1990 to 2010, reaching its highest value in 2010 (5.56) and then showing a downward trend. The Contagion Index showed a small increase from 1990 to 2000 and continued to decline by 5% from 2000 to 2020. According to the above findings, the research area's landscape heterogeneity increased from 1990 to 2020. The overall landscape structure in the study area gradually became fragmented, and the interference of human activities on the natural landscape increased.

**Table 7.** Landscape metrics in landscape level from 1990 to 2020.

| Year | PD | LPI | ED | AREA_MN | FRAC_MN | IJI | LSI | AI | SHDI | SHEI | CONTAG |
|---|---|---|---|---|---|---|---|---|---|---|---|
| 1990 | 0.52 | 5.22 | 45.47 | 192.27 | 1.1130 | 38.64 | 217.95 | 93.15 | 1.22 | 0.68 | 58.04 |
| 2000 | 0.52 | 5.28 | 45.71 | 191.89 | 1.1125 | 37.55 | 218.74 | 93.12 | 1.18 | 0.66 | 58.99 |
| 2010 | 0.53 | 5.56 | 45.20 | 189.40 | 1.1141 | 41.52 | 216.35 | 93.2 | 1.21 | 0.68 | 58.09 |
| 2020 | 0.65 | 4.73 | 47.20 | 152.96 | 1.1095 | 47.42 | 226.09 | 92.89 | 1.26 | 0.71 | 56.16 |

Note: PD indicates Patch Density; LPI is Largest Patch Index; ED is Edge Density; AR-EA_MN is Average Patch Area; FRAC_MN indicates Area-Weighted Mean Fractal Dimension Index; IJI refers to Interspersion and Juxtaposition Index; LSI refers to Landscape Shape Index; AI means Aggregation Index; SHDI is Shannon's Diversity Index; SHEI is Shannon's Evenness Index; CONTAG refers to Contagion Index.

The temporal changes of the landscape pattern index of each land type are as follows: (1) The Largest Patch Index and Aggregation Index of arable land show a growing and fluctuating pattern, but the Interspersion and Juxtaposition Index shows a reducing and then increasing trend. From 1990 to 2020, Patch Density increased by 13%, whereas

Average Patch Area declined by 14%. Edge Density, the Area-Weighted Mean Fractal Dimension Index, and Landscape Shape Index. Landscape Shape Index stayed essentially the same. (2) From 1990 to 2020, the Patch Density and Edge Density of forestland increased somewhat, whereas the Landscape Shape Index ascended dramatically by 20% from 2000 to 2020. Between 1990 and 2010, the Largest Patch Index remained essentially stable; however, between 2010 and 2020, it decreased by 56%. The Area-Weighted Mean Fractal Dimension Index and Aggregation Index had a consistent pattern. From 1990 to 2010, Average Patch Area was reasonably steady, but it fell by 13% from 2010 to 2020. The Interspersion and Juxtaposition Index exhibited a downward trend followed by an upward trend. (3) From 1990 to 2020, the Edge Density and Landscape Shape Index of grassland showed very minor variations and were relatively constant. In the last 30 years, the Interspersion and Juxtaposition Index has risen by 27%, whereas Area-Weighted Mean Fractal Dimension Index has remained essentially unaltered. However, the Largest Patch Index, Average Patch Area, and Aggregation Index all showed a tendency to initially increase, then drop. The Largest Patch Index and Average Patch Area had their highest and lowest levels in 2010, respectively, whereas the Aggregation Index had its highest value in 2000. In the last 30 years, Patch Density has shown a tendency to drop first and then increase, with the lowest number appearing in 2010. (4) The Largest Patch Index, Edge Density, Patch Density, Aggregation Index, and Area-Weighted Mean Fractal Dimension Index of the water area changed slightly and remained stable, but the Interspersion and Juxtaposition Index gradually rose. (5) The Largest Patch Index, Edge Density, the Landscape Shape Index, and Patch Density of construction land increased sharply from 2010 to 2020, whereas Average Patch Area, the Interspersion and Juxtaposition Index, and the Area-Weighted Mean Fractal Dimension Index increased rapidly from 2000 to 2010. The Aggregation Index change was relatively stable, with the lowest value in 1990 and the highest value in 2010. (6) Between 1990 and 2020, the Largest Patch Index and the Average Patch Area of unused land declined by 74% and 56%, respectively, whereas the Landscape Shape Index and Patch Density grew by 15% and 60%. The Aggregation Index of unused land fluctuated and was the lowest in 2000, and it had a large gap with other time nodes. The Landscape Shape Index, Edge Density, and the Area-Weighted Mean Fractal Dimension Index were all essentially the same.

The landscape pattern indexes of various land uses differ significantly (Table 8): (1) At each time node, the Patch Density of the arable land was the highest followed by the Patch Density of the grassland, and the Patch Density of the water area was the smallest. In addition, the Patch Density of all regions increased, and the changes in land-use patterns by humans gradually increased. (2) According to the Largest Patch Index and Average Patch Area results, the land-use types in the study area were mainly grassland and arable land. Except for construction land, the Largest Patch Index and Average Patch Area showed an increasing trend, whereas the Largest Patch Index and Average Patch Area of other land types showed a decreasing trend, especially for the unused land. They also showed that, except for building land, the fragmentation of other land categories grew in the study area. (3) The results of Edge Density and the Landscape Shape Index showed that the shapes of arable land and grassland were more complicated than other land types. (4) The Aggregation Index of each category was relatively high, between 89% and 97%. Among them, the Aggregation Index of unused land and grassland had changed greatly. (5) The Area-Weighted Mean Fractal Dimension Index of construction land was low, and the difference in the Area-Weighted Mean Fractal Dimension Indexes of other landscape types was small. (6) The Interspersion and Juxtaposition Index of construction land was the highest and increased sharply from 2000 to 2020, followed by the higher Interspersion and Juxtaposition Index of water areas with little change, whereas the Interspersion and Juxtaposition Index of arable land was the lowest.

**Table 8.** Landscape pattern index in class level from 1990 to 2020.

| Year | Class | PD | LPI | ED | LSI | AEREA_MN | FEAC_MN | IJI | AI |
|---|---|---|---|---|---|---|---|---|---|
| 1990 | Arable land | 0.23 | 4.14 | 37.32 | 294.10 | 156.07 | 1.125 | 28.59 | 92.23 |
| | Forestland | 0.07 | 0.18 | 3.38 | 88.83 | 48.96 | 1.088 | 66.22 | 92.24 |
| | Grassland | 0.13 | 5.22 | 40.99 | 298.48 | 331.43 | 1.126 | 38.95 | 92.74 |
| | Water | 0.01 | 0.29 | 1.24 | 56.80 | 90.9 | 1.111 | 67.48 | 92.20 |
| | Construction land | 0.03 | 0.01 | 0.56 | 43.65 | 11.2 | 1.052 | 54.62 | 88.65 |
| | Unused land | 0.05 | 2.86 | 7.46 | 87.67 | 346.23 | 1.099 | 50.54 | 96.59 |
| 2000 | Arable land | 0.23 | 3.86 | 37.39 | 294.7 | 155.46 | 1.125 | 26.12 | 93.15 |
| | Forestland | 0.07 | 0.18 | 3.49 | 89.95 | 49.39 | 1.088 | 63.54 | 92.21 |
| | Grassland | 0.12 | 5.28 | 41.78 | 292.66 | 387.50 | 1.126 | 39.52 | 95.95 |
| | Water | 0.01 | 0.29 | 1.21 | 54.86 | 96.37 | 1.113 | 68.41 | 92.30 |
| | Construction land | 0.03 | 0.02 | 0.58 | 43.20 | 12.47 | 1.053 | 55.38 | 92.38 |
| | Unused land | 0.06 | 1.14 | 6.96 | 92.19 | 229.10 | 1.099 | 44.37 | 89.42 |
| 2010 | Arable land | 0.23 | 4.58 | 35.80 | 287.54 | 148.20 | 1.128 | 27.66 | 92.26 |
| | Forestland | 0.08 | 0.18 | 4.49 | 104.59 | 48.65 | 1.093 | 60.98 | 91.88 |
| | Grassland | 0.11 | 5.56 | 41.00 | 285.24 | 441.86 | 1.123 | 44.77 | 93.37 |
| | Water | 0.01 | 0.14 | 1.21 | 56.30 | 91.30 | 1.12 | 72.88 | 91.94 |
| | Construction land | 0.03 | 0.06 | 0.58 | 46.30 | 30.62 | 1.066 | 77.36 | 92.84 |
| | Unused land | 0.06 | 1.14 | 6.96 | 93.81 | 215.68 | 1.100 | 47.47 | 95.78 |
| 2020 | Arable land | 0.26 | 4.45 | 36.48 | 292.97 | 134.62 | 1.125 | 32.33 | 93.06 |
| | Forestland | 0.09 | 0.08 | 4.51 | 106.71 | 43.87 | 1.092 | 65.43 | 92.11 |
| | Grassland | 0.15 | 4.73 | 41.68 | 294.27 | 301.95 | 1.115 | 49.49 | 91.59 |
| | Water | 0.02 | 0.13 | 1.30 | 61.08 | 60.59 | 1.108 | 75.18 | 92.08 |
| | Construction land | 0.06 | 0.47 | 3.16 | 86.72 | 52.66 | 1.074 | 77.19 | 91.27 |
| | Unused land | 0.08 | 0.75 | 7.25 | 100.69 | 151.40 | 1.095 | 58.33 | 95.36 |

### 3.3. Spatiotemporal Change in Habitat Quality

The habitat quality indexes for 1990, 2000, 2010, and 2020 were calculated using the habitat quality module of the InVEST model. The habitat quality index is a number between 0 and 1 that indicates how good a habitat is. Using the natural breaks method (Jenks), the habitat quality value was divided into four levels: low habitat quality (0.0–0.2), moderate habitat quality (0.2–0.4), good habitat quality (0.4–0.6), and high habitat quality (0.6–1.0).

The average habitat quality values in 1990, 2000, 2010, and 2020 are 0.358, 0.347, 0.365, and 0.357, respectively (Table 9). The average habitat quality value first decreased, then increased, and then decreased, but the change range was relatively slight. From the perspective of the habitat quality classification level, the good habitat quality area had the biggest proportion, followed by the moderate habitat quality area, the poor habitat quality area, and finally the high habitat quality area. The sum of the proportions of the good habitat quality area and the moderate habitat quality area accounts for more than 80% of the research region, indicating that the habitat quality of the study area was at a relatively good level. Regarding temporal variation, the low habitat quality area decreased by 3% between 1990 and 2020. The proportion of moderate habitat quality area increased sharply by 14% from 1990 to 2000 and decreased by 11% from 2000 to 2020. However, the good habitat quality area decreased by 10% between 1990 and 2000 and increased by 9% between 2000 and 2020. The high habitat quality area increased by 1% between 2000 and 2010, and it remained the same between 1990 and 2000 and between 2010 and 2020. In general, the average habitat quality in the study area has slightly improved, but not significantly.

**Table 9.** Area and ratio of each habitat quality level from 1990 to 2020.

| HQ Level | Value Interval | 1990 | | 2000 | | 2010 | | 2020 | |
|---|---|---|---|---|---|---|---|---|---|
| | | Km$^2$ | % | Km$^2$ | % | Km$^2$ | % | Km$^2$ | % |
| Low | 0–0.2 | 6092.62 | 17 | 4828.64 | 13 | 4765.88 | 13 | 5274.82 | 14 |
| Moderate | 0.2–0.4 | 9579.31 | 27 | 14,564.06 | 41 | 10,924.88 | 31 | 10,658.70 | 30 |
| Good | 0.4–0.6 | 19,168.11 | 54 | 15,542.94 | 44 | 18,913.10 | 53 | 18,732.28 | 53 |
| High | 0.6–1 | 789.19 | 2 | 693.59 | 2 | 1024.47 | 3 | 963.43 | 3 |
| Mean | | 0.358 | | 0.347 | | 0.365 | | 0.357 | |

The habitat quality of the study area had significant spatial-distribution changes, presenting staggered distribution characteristics (Figure 4). The high-habitat-quality regions were mainly fragmented and scattered in the waters and forests of the study area. Forests and waters provide excellent habitats for most species. The good-habitat-quality regions were distributed in the northeast and south, and the distribution characteristics were related to the distribution of grassland. The moderate-habitat-quality regions were concentrated in the southwest and central, and the main land type was arable land. Low-habitat-quality regions were mainly distributed in the western and central parts of the study area, and the main land types were construction land and unused land.

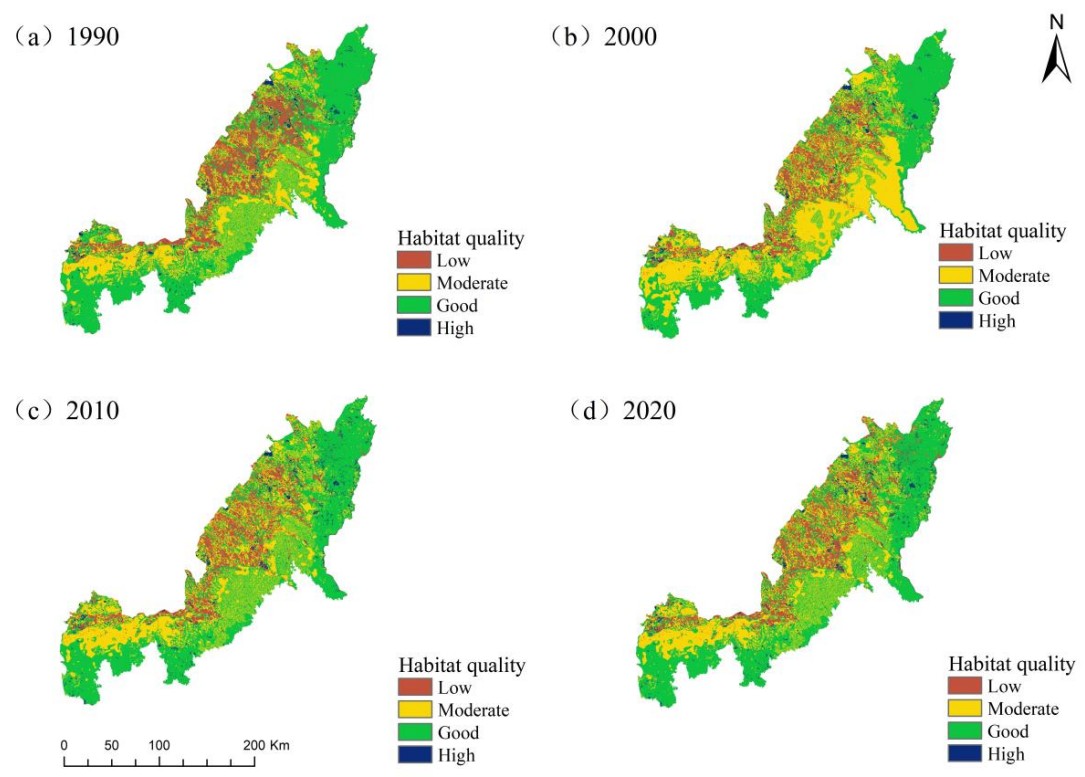

**Figure 4.** Spatial distribution and change in habitat quality in the study area: (**a**) 1990; (**b**) 2000; (**c**) 2010; (**d**) 2020.

The habitat-quality difference in the study area was calculated using the raster calculator tool of the ArcGIS 10.2. From 1990 to 2020, we found a spatial depiction of the dynamic increase and reduction in habitat quality in northern Shaanxi's agro-pastoral ecotone, and we investigated the spatial aspects of habitat quality further (Figure 5). The habitat quality of most regions of the research area remained steady from 1990 to 2000, accounting for 84% of the study area's total area. The areas with significantly improved habitat quality were mainly concentrated in the north and west, accounting for 1% of the total area of the study area, and areas with slightly improved habitats were mainly in the east and

north, accounting for 8% of the total area of the study area. The study area's slightly degraded habitats were mostly concentrated in the southwest, accounting for 5% of the total area. Significantly degraded habitats were mostly in the northeast and central parts of the study area, accounting for 2% of the total area. From 1990 to 2000, there was slight habitat degradation in Dingbian County's southern and northern parts, in eastern Hengshan County, in eastern Yuyang County, in southern Shenmu County, and in the majority of Jia County. The northern part of Dingbian County had the most significant habitat degradation. Significant habitat degradation was distributed primarily in the northern part of Dingbian County. The areas with slightly improved habitat quality were mostly found in Yuyang District and in Shenmu County's northwestern area. From 2000 to 2010, habitat improvement areas increased significantly. Habitat quality in the south and north of Dingbian County, in the east of Hengshan County and Yuyang District, in the south of Shenmu County, and in most areas of Jia County improved slightly. The areas with degraded habitats were mainly scattered in the central and northern parts of Shenmu City. From 2010 to 2020, the habitat-quality-improvement area was mainly in the west of the study area. The land-use type in the west of the study area was mainly composed of grassland and sand in unused land. Habitat-degradation areas were mainly found in the north and middle of Shenmu County, in the middle of Yuyang District, in the west of Hengshan County, and in the north of Jingbian County and Dingbian County. These fragmented habitat-degradation areas present a belt-like spatial-distribution pattern, which is similar to the characteristics of the spatial expansion of construction land from 2010 to 2020.

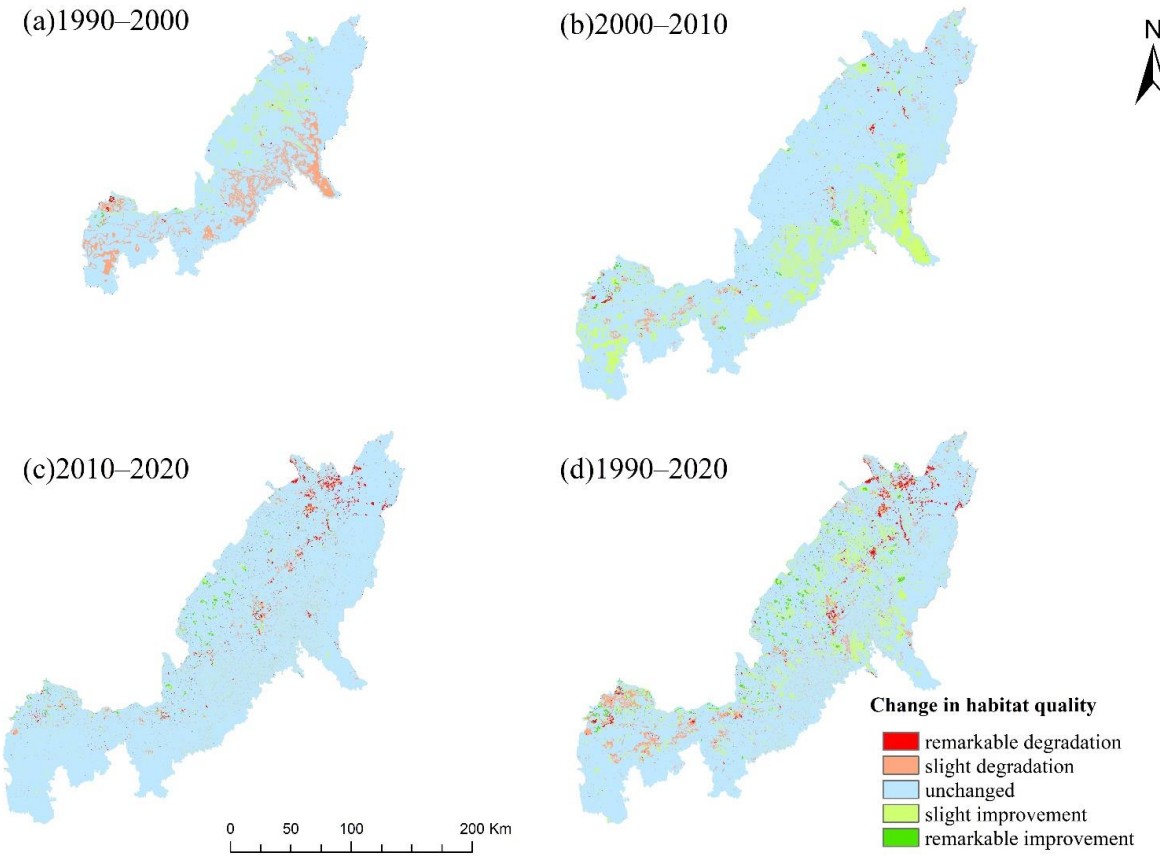

**Figure 5.** Spatial change in habitat quality in study area from 1990 to 2020: (**a**) 1990–2000; (**b**) 2000–2010; (**c**) 2010–2020; and (**d**) 1990–2020.

The spatial distribution of habitat quality was investigated using global Moran's I and hot spot analysis [15]. The study area's habitat quality distribution was examined using Global Moran's I. In 1990, 2000, 2010, and 2020, the Global Moran's I in the research region

was 0.199, 0.289, 0.202, and 0.258, respectively, and the p-values were all 0. This shows that habitat quality had a spatial agglomeration effect (Figure 6). Then, the spatial aggregation and distribution characteristics of habitat quality were explored using hot spot analysis. Taking 2020 as an example, the general geographical distribution of habitat quality in the research region was "high in the east and south, and low in the west and north". The hot spots were mostly concentrated in the forest–grass interlaced area in the northeast and south of the study area, with a few scattered in the forestlands and waters in the north and center. The cold spots were mostly distributed on sandy land, construction land, and a portion of arable land.

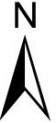
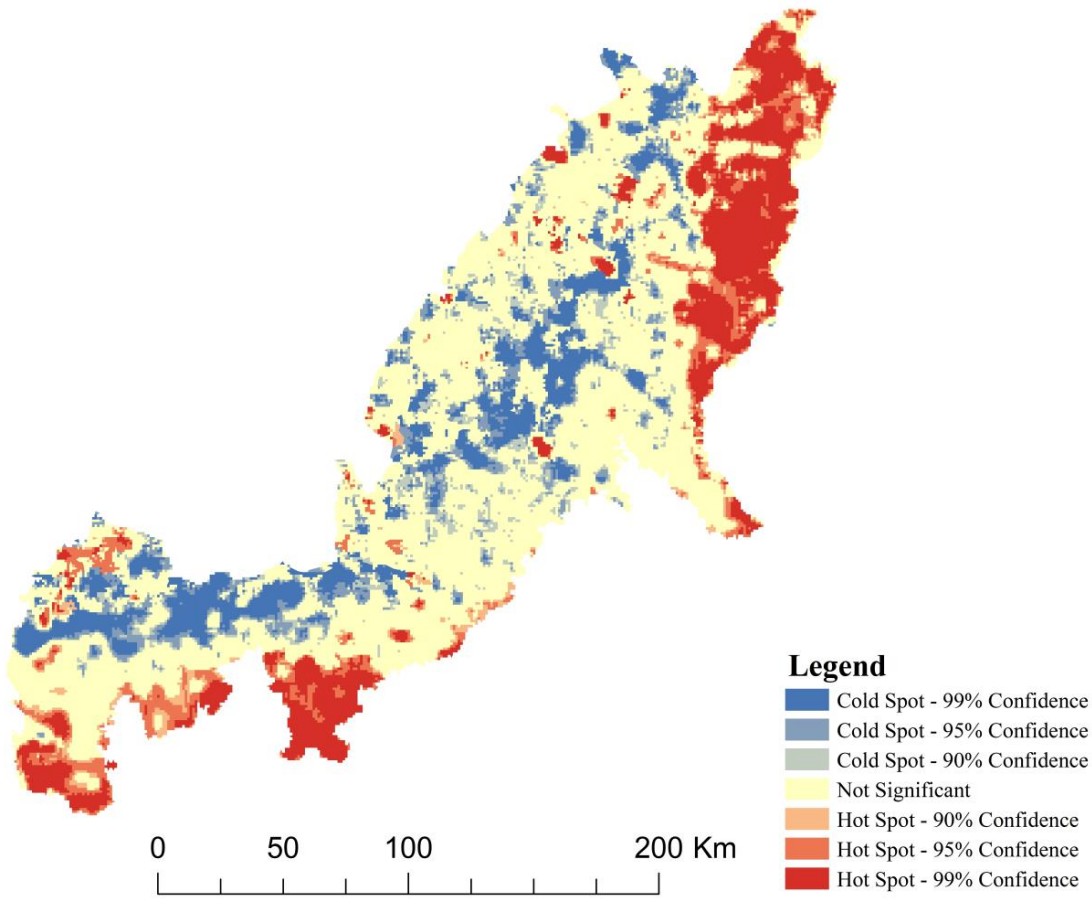

**Figure 6.** The spatial distribution of hot spots and cold spots of habitat quality in 2020.

### 3.4. Changes in Habitat Quality Characteristics of Different Land Types

Land use had a major effect on regional habitat quality, and changes in land use led to habitat-quality changes [13], potentially leading to increased habitat fragmentation. Human activities alter spatial patterns of land use, leading to habitat changes and, as a result, to biodiversity change. To better understand the impact of land-use changes on habitat quality, we conducted a statistical analysis of the habitat quality level of different land-use types in the study region in 1990, 2000, 2010, and 2020 (Table 10). The habitat quality levels of different land-use types showed certain changes during the four research periods. The arable land was mainly of good-level quality overall. The habitat quality of arable land was inferior in 2000, and the proportion of good-level areas declined sharply. The dominant level of habitat quality of forestland was good-level and high-level, but the ratio of good and high levels continued to decline from 1990 to 2020. This indicates that the habitat quality of forestland degraded. The habitat-quality level of the water area was also good-level and high-level. From 1990 to 2020, the proportion of high levels continued to decline. The habitat-quality level of grassland was mostly moderate-level

and good-level. The proportion of good levels decreased before increasing, whereas the changing trend of moderate-level was the opposite. The construction land was distributed in low-level habitat-quality areas. The habitat quality of construction land was low-level. The main habitat-quality level of unused land was low-level, but the proportion slightly decreased during the study period, and the habitat quality of unused land slightly improved. The spatial distribution of habitat quality was found to be highly related to land-use types.

**Table 10.** Habitat-quality level change in different land-use types in study area from 1990 to 2020.

| Land-Use Type | Year | Low | Moderate | Good | High |
|---|---|---|---|---|---|
| Arable Land | 1990 | 0% | 19% | 81% | 0% |
| | 2000 | 0% | 42% | 58% | 0% |
| | 2010 | 0% | 19% | 81% | 0% |
| | 2020 | 0% | 19% | 81% | 0% |
| Forestland | 1990 | 0% | 10% | 46% | 44% |
| | 2000 | 0% | 16% | 47% | 37% |
| | 2010 | 0% | 27% | 36% | 37% |
| | 2020 | 0% | 27% | 38% | 35% |
| Grassland | 1990 | 0% | 46% | 53% | 1% |
| | 2000 | 0% | 55% | 45% | 0% |
| | 2010 | 0% | 49% | 49% | 2% |
| | 2020 | 0% | 48% | 50% | 2% |
| Water | 1990 | 1% | 10% | 52% | 37% |
| | 2000 | 4% | 13% | 51% | 32% |
| | 2010 | 2% | 8% | 60% | 30% |
| | 2020 | 1% | 11% | 57% | 31% |
| Construction Land | 1990 | 100% | 0% | 0% | 0% |
| | 2000 | 100% | 0% | 0% | 0% |
| | 2010 | 100% | 0% | 0% | 0% |
| | 2020 | 100% | 0% | 0% | 0% |
| Unused Land | 1990 | 100% | 0% | 0% | 0% |
| | 2000 | 99% | 1% | 0% | 0% |
| | 2010 | 98% | 2% | 0% | 0% |
| | 2020 | 98% | 2% | 0% | 0% |

*3.5. Analysis of the Driving Force of Habitat Quality Changes Based on the Geo-Detector*

To study the driving force of land use and other factors on habitat quality quantitatively, factor detectors and interactive detectors were used to calculate the driving forces of each factor. Table 11 shows the driving force of various geographical factors on habitat quality. The p-values of all factors were <0.05, passing the significance test. The order of q values was: X5 > X6 > X3 > X7 > X2 > X8 > X4 > X1. The driving force of the land-use factor was 0.579, which was the strongest driving force for habitat quality. Changes in land-use patterns significantly affected the spatial differentiation characteristics of habitat quality. In addition, vegetation coverage and precipitation also had strong driving forces for habitat quality, which were 0.182 and 0.114, respectively. These three factors were the significant driving factors for habitat quality. Secondly, slope (0.098), GDP (0.075), and elevation (0.046) had a certain impact on habitat quality. The single-factor driving force of population density on habitat quality was the smallest, at only 0.003.

**Table 11.** The results of single-factor detection.

| Driving Factors. | X1 | X2 | X3 | X4 | X5 | X6 | X7 | X8 |
|---|---|---|---|---|---|---|---|---|
| Driving Force (q) | 0.003 | 0.075 | 0.114 | 0.020 | 0.579 | 0.182 | 0.098 | 0.046 |
| *p*-value | 0.039 | 0 | 0 | 0 | 0 | 0 | 0 | 0 |

Note: X1 is population density, X2 is GDP per square kilometer, X3 is annual precipitation, X4 is annual temperature, X5 is land use, X6 is NDVI, X7 is slope, and X8 is elevation.

We used interactive detectors to detect the relationship among driving factors that affect habitat quality. From the interaction detector's results (Table 12), the interactions were both bi-factor enhancement and nonlinear enhancement. The combinations of precipitation and land use, the combination of GDP and land use, the combination of land use and slope, the combination of land use and NDVI, and the combination of land use and elevation were 0.67, 0.63, 0.61, 0.60, and 0.60, respectively. These driving forces exceed 60%. The driving force of the spatial differentiation of mass is the highest. The second is the combination of temperature and land use (0.59) followed by the combination of population density and land use (0.58). The interaction between land use and others was obviously stronger than the interaction between other factors. This also reflects that land use was the main driving factor for habitat quality. In addition, the combination of precipitation and NDVI (0.24), GDP and NDVI (0.23), and temperature and NDVI (0.20) were all greater than 0.2, indicating that the interactions between NDVI and other factors were also significant. The bi-factor interactions had different levels of enhancement, indicating that the interaction of factors had a stronger impact on habitat quality.

**Table 12.** The results of interactive detection.

| Interaction | Influence | Interaction | Influence |
|---|---|---|---|
| X1 ∩ X2 (0.10) | Nonlinear Enhancement | X3 ∩ X5 (0.67) | Double-Factor Enhancement |
| X1 ∩ X3 (0.12) | Double-Factor Enhancement | X3 ∩ X6 (0.24) | Double-Factor Enhancement |
| X1 ∩ X4 (0.02) | Nonlinear Enhancement | X3 ∩ X7 (0.15) | Double-Factor Enhancement |
| X1 ∩ X5 (0.58) | Nonlinear Enhancement | X3 ∩ X8 (0.16) | Double-Factor Enhancement |
| X1 ∩ X6 (0.19) | Nonlinear Enhancement | X4 ∩ X5 (0.59) | Double-Factor Enhancement |
| X1 ∩ X7 (0.05) | Nonlinear Enhancement | X4 ∩ X6 (0.20) | Double-Factor Enhancement |
| X1 ∩ X8 (0.10) | Nonlinear Enhancement- | X4 ∩ X7 (0.06) | Double-Factor Enhancement |
| X2 ∩ X3 (0.15) | Nonlinear Enhancement | X4 ∩ X8 (0.11) | Double-Factor Enhancement |
| X2 ∩ X4 (0.10) | Double-Factor Enhancement | X5 ∩ X6 (0.60) | Double-Factor Enhancement |
| X2 ∩ X5 (0.63) | Double-Factor Enhancement | X5 ∩ X7 (0.61) | Double-Factor Enhancement |
| X2 ∩ X6 (0.23) | Double-Factor Enhancement | X5 ∩ X8 (0.60) | Double-Factor Enhancement |
| X2 ∩ X7 (0.12) | Nonlinear Enhancement | X6 ∩ X7 (0.05) | Double-Factor Enhancement |
| X2 ∩ X8 (0.15) | Double-Factor Enhancement | X6 ∩ X8 (0.12) | Double-Factor Enhancement |
| X3 ∩ X4 (0.12) | Nonlinear Enhancement | X7 ∩ X8 (0.10) | Double-Factor Enhancement |

## 4. Discussion

In this study, the habitat-quality pattern changes of the agro-pastoral ecotone in northern Shaanxi from 1990 to 2020 and their driving factors were evaluated by the habitat quality module of the InVEST model and Geo-detector. The research results play meaningful and significant roles in the conservation of biodiversity and in the construction of ecological civilizations in semi-arid and arid regions.

### 4.1. The Relationship of Land-Use Change and Habitat Quality

This study shows a slight fluctuation trend in mean habitat quality between 1990 and 2020. From the perspective of changes in habitat-quality level, the area of improved habitat quality was slightly greater than the area of habitat-quality degradation, indicating that the habitat quality in the study area slightly improved. The habitat quality of the study area had obvious spatial heterogeneity, showing a "low in the west, high in the east", spatial-distribution pattern. The areas with high and good habitat quality were mainly forestland, water area, and grassland. Construction land, unused land, and arable land were mainly in areas with moderate and low habitat quality, which is consistent with the research results of Li et al. [13] and Yang et al. [15]. From 1990 to 2020 in the study area, because of the Three-North Shelterbelt Project, the ecological restoration project, and the Grain for Green Project in the Mu Us Sandy Land, forestland and grassland in the study area increased, which helped to enhance habitat quality. However, against the backdrop of increasing population, socioeconomic development, and urbanization, construction land increased by nearly seven times in the past 30 years, especially construction scale and land development intensity in the urban areas of the Yuyang district and counties in the middle of the research area. The expansion of construction land affected the quality of habitats, gradually connecting into strips, forming a new threat source, and the surrounding habitat was squeezed and disturbed.

### 4.2. Driving Factors of Habitat Quality

Land use was a dominant driving force affecting habitat quality, which is consistent with the results of Yang et al. [42]. Empirical studies [43–46] show that forestland often has high species richness, which can provide more supply and support services for species survival. Water has good survival conditions for aquatic organisms. Grassland has lower biodiversity than forestland. Unused land is relatively harsh, where only a few species can survive, and its biodiversity is low. Arable land and construction land have reduced biodiversity in comparison to their original habitats due to human influence. Different from the results of Zhang et al. in Liulin County, Shanxi Province of China [47], our results show that the driving force of precipitation and vegetation cover on the habitat quality of the agro-pastoral ecotone in northern Shaanxi is stronger than the driving force of elevation and slope. Because the study area is in the transition zone between semi-arid areas, precipitation is significant for the local ecological environment. Precipitation with uneven spatial distribution resulted in significant spatial heterogeneity of vegetation coverage in the study area, which affected habitat quality. In addition, slope and elevation affected the site selection of human life and production activities, which in turn drove the spatial differentiation of habitat quality [48]. The GDP and population-density factors belonging to socioeconomic factors were less strongly driven by habitat quality, perhaps not sufficiently significant at the scale of the study area. The effects of the interactions among the factors on habitat quality were enhanced, indicating that bi-factor coupling had a stronger driving effect than a single factor. The interaction driving force between land use and other factors was significantly greater than the interaction driving force between other factors, further demonstrating that land use was a main factor in habitat-quality change. Natural factors and socioeconomic factors jointly drove the changes in habitat quality. The correlations among the influencing factors of habitat-quality variations are complex and interactive.

*4.3. Implication for Habitat-Quality Conservation*

Regional habitat quality is affected by many factors, so habitat quality improvement and ecological protection should be based on local regional resource endowments and should follow the concept of comprehensive, coordinated, and sustainable development. Administration should formulate differentiated ecological protection strategies. For urban land development, authorities should increase the intensive utilization of construction land, limit the development boundaries of urban land, and increase the greening rate inside urban land. Authorities should pay special attention to the conservation of arable land and to the development of ecological agriculture regarding agricultural production [49]. For ecological protection, authorities should improve the natural reserve system by constructing biodiversity conservation networks and ecological corridors [50]. The vegetation ecological restoration project should be carried out according to local conditions. Authorities have carried out ecological restoration projects, such as the Grain for Green program and the Three-North Shelterbelt project. However, regarding artificial vegetation restoration, management should avoid planting a single species plant and should build a biological community [51] with high species richness. Due to the shortage of water resources in the study area, we should avoid planting trees blindly. It is necessary to combine local water resources for afforestation in areas that are suitable for afforestation and for grass planting in areas that are suitable for grass planting. The government's attention should be paid to the application of native plants in the restoration of artificial vegetation to improve the species diversity of artificial vegetation and to build high species richness in communities dominated by shrubs and grasses. Thereby, habitat quality and the biodiversity-maintenance function of regional ecosystems can be improved. Therefore, it is necessary to implement reasonable and effective territorial space planning to achieve regional sustainable development.

*4.4. Limitations and Prospective*

There are some weaknesses in this study that need to be improved: (1) Many factors are required by the InVEST model, and the lack of certainty and sensitivity of these parameters can influence the model's results. Due to the lack of standard parameter settings for InVEST, the relevant parameters used in this study are adjusted based on model manuals, the literature, and expert experience, which is subjective and may lead to findings that differ from those reported [52]. (2) This study calculates habitat quality based on linear threat sources and area threat sources but does not consider the impact of point threat sources such as pollutant emissions [53]. In the future, we should strengthen point-source pollution data and consider the impact of specific human activities on habitat quality, as well as supplement and optimize model parameters to improve model accuracy [54]. (3) The detected factors' raster data resolution is limited, and there are certain deviations when using fishing nets to sample, which may lead to deviations in the driving force results obtained by inputting the Geo-detector model [54]. (4) There may be more factors affecting heterogeneity in habitat quality, such as the Normalized Difference Water Index (NDWI), the range of nature conservation areas, etc. However, we failed to obtain these data, and they should be improved in the next study.

## 5. Conclusions

Exploring spatiotemporal changes and the driving mechanisms of habitat quality is significant for biodiversity conservation in ecosystems. This paper uses the InVEST model and Geo-detector to explore the temporal and spatial changes and the driving forces of the agro-pastoral ecotone in northern Shaanxi from 1990 to 2020. The results show the following: (1) From 1990 to 2020, arable land and grassland were the main land types in the study area. There were many mutual transformations between arable land and grassland. Forestland continued to increase by 24%, whereas water area decreased by 8%. The unused land was reduced by 30% and mainly flowed to grassland. There are many other land-use types that were transformed into construction land, and construction

land increased by 727% compared with the base period. (2) From 1990 to 2020, landscape heterogeneity in the study area increased, the overall landscape structure in the study area has been gradually fragmenting, and the interference of human activities on natural land has increased. There were large differences in the landscape pattern index of different land types. The fragmentation of forestland, arable land, and grassland was enhanced, and the shape of land-use patches became more complex. Habitat types such as woodland, water, and grassland in the study area are relatively broken. These fragmented ecosystems are vulnerable to human activities, and biodiversity is vulnerable to damage. However, the fragmentation degree of construction land has decreased, the degree of aggregation has increased, and its spatial distribution has tended to be concentrated. (3) Between 1990 and 2020, the mean habitat quality of the agro-pastoral ecotone in northern Shaanxi experienced a process of first decreasing quality, then increasing quality, and finally decreasing again. The spatial distribution of overall habitat quality in the study area was "high in the east and low in the west". (4) Land use was the main driving factor of habitat quality, followed by natural factors such as vegetation coverage and precipitation. Socioeconomic factors such as population density and GDP were weakly driven. In addition, two-factor interactions were enhanced to different levels.

**Author Contributions:** Conceptualization, G.C. and Y.Z.; Formal analysis, W.J. and B.P.; Funding acquisition, Y.Z.; Methodology, G.C.; Software, G.C. and F.S.; Visualization, Z.L. and H.Z.; Writing—original draft preparation, G.C.; Writing—review and editing, C.H. and M.L. All authors have read and agreed to the published version of the manuscript.

**Funding:** This study was funded by the Fundamental Research Funds for the Central Universities, CHD (300102291507), and by the Program of Introducing Talents of Discipline to Universities (B08039).

**Institutional Review Board Statement:** Not applicable.

**Informed Consent Statement:** Not applicable.

**Data Availability Statement:** Not applicable.

**Acknowledgments:** The authors are grateful to reviewers for their valuable comments and suggestions.

**Conflicts of Interest:** The authors declare no conflict of interest.

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
