# Peer review of "Study of Spatiotemporal Changes and Driving Factors of Habitat Quality: A Case Study of the Agro-Pastoral Ecotone in Northern Shaanxi, China"

_sustainability, doi:10.3390/su14095141_

Round 1

Reviewer 1 Report

Summary and general comments

The manuscript in review investigates habitat quality and drivers of change in habitat quality in the agro-pastoral ecotone of the northern Shaanxi region in China. The authors used the Habitat Quality module of Invest software and run Geodetector model using a spatial dataset. The dataset covers land use data from 1990, 2000, 2010, and 2020 to detect spatial-temporal changes in habitat quality and landscape patterns in the region. The novel aspect of the study is that it aims to understand the contribution of different factors such as GDP, population density, vegetation, precipitation, temperature, slope, and elevation, as standalone and in combination.  

The manuscript has a clear structure, and it is intriguing the reader. The methodology is accurate and replicable to a certain extent. The tables and figures are appropriate and sufficient, but the quality of the images should be improved. Statistical methods are correctly applied.

Abstract

In the abstract, landscape fragmentation analysis is not mentioned alongside habitat quality and geodetector analysis. This section should also introduce the layers used in geodetector analysis.

Introduction

This section briefly introduces a literature review related to the issue. However, there is a need for a more in-depth review of the literature. Citing of previous studies should critically elaborate what methods/factors/ layers etc. Used in previous studies that lead to your selection of methods and variables. However, highlights the general gap of focusing on the impact of different factors.

Line 80: landscape pattern index can be introduced in addition to Invest.

Methods

Landscape pattern

Line 126: Fragstat- Initials of the software, who developed and when? Also, the landscape pattern index and its calculation should be briefly elaborated in the literature review. In addition, the reason behind why you choose 11 landscape pattern indexes should be shortly introduced along with the specific characteristics and meanings of these indices.  

Although the layers of precipitation, temperature, GDP, Ndvi, and population density are ready to use spatial data retrieved from a data center, the details of these layers- metadata- should be briefly introduced. For example, how GDP, precipitation, and temperature maps are produced. In addition, what do you think about the impact of different resolutions on the results of the analysis? Maybe this point could be included in the limitations of the study section.

Results

Line 351- Moran’s I and hotspot analysis should be introduced in the methodology section.

A discussion part can be integrated as a subchapter from line 422.

Line 486 points to the limitations of the study. Maybe limitations and further opportunities in using different indices can be discussed in this part. A limited discussion on whether different indices such as NDWI, a protected areas layer, etc. could yield different results or not can be included. In addition, understanding habitat quality in water bodies might require the integration of indices as used in vegetation. You can elaborate on these dimensions if you find them suitable.

Overall, the results section formally introduces the results and the discussion part can integrate context-specific local and national policy and planning drivers.

The conclusion section is adequate, yet some insights from the discussion section could be integrated with further areas of research and policy implementation.

Figures

All figures should be improved in terms of resolution and quality.

Figure 1- coordinates are not visible

Figure 2- All small images should be enlarged for a better view. The values should be visible.

Figure 3,4,5,6,7- Image resolution should be improved.

Table 9- the details of the factors in the table fall to the next page. This information could be on the same page as with the table to have fluent reading.

Author Response

Response to Reviewer 1 Comments

Thanks very much for taking your time to review this manuscript. We really appreciate all your comments and suggestions!

Point 1: In the abstract, landscape fragmentation analysis is not mentioned alongside habitat quality and geodetector analysis. This section should also introduce the layers used in geodetector analysis.

Response 1: We modify the abstract to add a brief introduction to the landscape pattern and geodetector detected layers. Modified as follows:

Abstract: Habitat quality is a key indicator for assessing biodiversity-maintenance functions of ecosystem services. The issue of habitat quality in semi-arid and arid areas has been getting serious, but there are few deep investigations of habitat quality in these regions, such as to study the temporal and spatial changes of habitat quality and its driving forces. This study focuses on the agro-pastoral ecotone of northern Shaanxi which is a transitional zone between semi-arid and arid areas with vulnerable biodiversity. By using the Fragstats software, InVEST model, and the Geo-detector model, we analyzed land use data collected from 1990, 2000, 2010, and 2020, explored the landscape pattern index, the spatial and temporal variation of habitat quality and the influence of its drivers. GDP, Population density, precipitation, temperature, land use, NDVI, elevation and Slope were detected by Geo-detector. The research results showed that: (1) Arable land and grassland were the dominant land types during 1990- 2020, and there was significant mutual circulation between arable land and grassland. The forest area increased by 24%. Many other land use types were transformed into construction land, and the construction land increased by 727% compared with the base period. (2) The landscape heterogeneity was increased in the study region, shown by fractured structure of the overall landscape and aggravated human disturbance of landscape. (3) The average habitat quality went through a trend of oscillation. In terms of spatial distribution, habitat quality was higher in the east than in the west. (4) The influencing factors of habitat quality monitored by Geo-detectors showed that the driving force of land use on habitat quality was the strongest, followed by precipitation and vegetation coverage. Elevation, slope, GDP and population density had the least influence on habitat quality. The bi-factor interaction enhanced the habitat quality to different levels. This study is critical to the conservation of biodiversity and the ecological civilization construction in the arid and semi-arid regions.

Point 2: This section briefly introduces a literature review related to the issue. However, there is a need for a more in-depth review of the literature. Citing of previous studies should critically elaborate what methods/factors/ layers etc. Used in previous studies that lead to your selection of methods and variables. However, highlights the general gap of focusing on the impact of different factors.

Response 2: We revised the introduction. We updated some text of the methods of driving force for habitat quality.

Point 3:Line 126: Fragstats- Initials of the software, who developed and when? Also, the landscape pattern index and its calculation should be briefly elaborated in the literature review. In addition, the reason behind why you choose 11 landscape pattern indexes should be shortly introduced along with the specific characteristics and meanings of these indices.  

Response 3: We added a brief introduction to the Fragstats. Modified as follows: Fragstats is a spatial pattern tool for quantifying the structure of landscapes. It is used to calculate metrics of landscape[27]. This software was developed by University of Mas-sachusetts in 1995.

Point 4:Although the layers of precipitation, temperature, GDP, Ndvi, and population density are ready to use spatial data retrieved from a data center, the details of these layers- metadata- should be briefly introduced. For example, how GDP, precipitation, and temperature maps are produced. In addition, what do you think about the impact of different resolutions on the results of the analysis? Maybe this point could be included in the limitations of the study section.

Response 4:We added the information about data. Specifically, temperature distribution raster data and precipitation distribution raster data were generated using ANUSPLIN interpolation software. NDVI spatial distribution dataset was based on SPOT/VEGETATION PROBA-V 1 KM PRODUCTS ten-day 1km vegetation index data, based on monthly data using the maximum synthesis method to generate Annual vegetation index dataset since 1998. GDP spatial distribution raster data was based on the national GDP statistics data of counties, and comprehensively considers multiple factors such as land use type, night light brightness, and density of settlements that are closely related to human economic activities. The method spreads the GDP data with the administrative area as the basic statistical unit to the grid unit, to realize the spatialization of GDP.

Point 5:Moran’s I and hotspot analysis should be introduced in the methodology section.

Response 5:We updated the content of Moran’s I and hotspot analysis in the methodology section.

Point 6:A discussion part can be integrated as a subchapter from line 422

Response 6:We integrated as a sub-chapter for discussion.

Point 7: Line 486 points to the limitations of the study. Maybe limitations and further opportunities in using different indices can be discussed in this part. A limited discussion on whether different indices such as NDWI, a protected areas layer, etc. could yield different results or not can be included. In addition, understanding habitat quality in water bodies might require the integration of indices as used in vegetation. You can elaborate on these dimensions if you find them suitable.

Response 7:In the Discussion section, we show that the lack of detection of some other habitat quality influencing factors has limitations to the acquisition of drivers of habitat quality. It should be perfected in the next research step.

Point 8: The conclusion section is adequate, yet some insights from the discussion section could be integrated with further areas of research and policy implementation.

Response 8:We revised the discussion part based on the background of local ecological projects.

Point 9: All figures should be improved in terms of resolution and quality.

Response 9: We used improved the resolution and quality of these figures.

Reviewer 2 Report

The manuscript titled "Study of spatiotemporal changes and driving factors .....in Northern China.   The quality of the manuscript is in general publishable quality. But the following are the areas for improvement.    If this study does comparative examination of habitat quality changes between arid and semiarid regions, I advise giving the title accordingly. It is based on the fact that there are thousands of studies in spatiotemporal changes of habitat quality.    Abstract The abstract can be substantiated. I suggest focusing on the differences of habitat quality dynamics between arid and semiarid regions.   Introduction: The authors missed stating the importance of better understanding of habitat quality dynamics such as "The importance of understanding on the human impacts are increasing with rapid changes on land use activities".   ​Methods. I suggest defining land pattern index at page 4. There is scope of improving language from line 150-155.   Table 4: The joining symbol "inverted U" required defining in a note under the table ​ Figure 2: Index are not clear   Results   I suggest to interprate all the results of the habitat change in percentage with areas in parentheses. I advise uses of meaningful key words instead of abbreviations in result section 3.1 and after. Use of the abbreviation makes the readers confusing to understand. Figure 4 : Unclear leveling of the graphs Figure 6: unclear leveling.   Instead of comparing between provisions please use the term arid and semi arids where appropriate to describe the habitat changes.    Conclusions Suggest drawing the conclusion of the habitat dynamic based on regional ecological characteristics.

Author Response

We would like to thank you for your careful reading, helpful comments, and constructive suggestions, which has significantly improved the presentation of our manuscript.

This manuscript is a resubmission of an earlier submission. The following is a list of the peer review reports and author responses from that submission.